# Psychosocial impact of COVID-19 pandemic on front-line healthcare workers in Sierra Leone: an explorative qualitative study

Sia Morenike Tengbe ,[1] Ibrahim Franklyn Kamara ,[2] Desta B Ali ,[1] Fanny F Koroma,[1] Stephen Sevalie,[3,4] Laura Dean,[5] Sally Theobald[5]

¹Sierra Leone Ministry of Health and Sanitation, Freetown, Sierra Leone
²Emergency Preparedness and Response Cluster, World Health Organisation Country Office for Sierra Leone, Freetown, Sierra Leone
³34 Military Hospital, Wilberforce, Freetown, Sierra Leone
⁴Case Management Pillar, National COVID-19 Emergency Response Centre, Freetown, Sierra Leone
⁵International Public Health, Liverpool School of Tropical Medicine, Liverpool, UK

**Correspondence to**
Dr Sia Morenike Tengbe;
siamoreniketengbe@outlook.com

## ABSTRACT

**Introduction** The COVID-19 pandemic has wide-reaching health and non-health consequences, especially on mental health and psychosocial well-being. Healthcare workers involved in COVID-19 patient care are particularly vulnerable to psychosocial distress due to increased pressure on healthcare systems. We explored the psychosocial experiences of front-line healthcare workers during the COVID-19 pandemic in Sierra Leone.

**Methods** This qualitative study used purposive sampling to recruit 13 healthcare workers from different cadres across 5 designated COVID-19 treatment centres in Freetown, Sierra Leone. In-depth interviews were conducted remotely in July and August 2020, transcribed verbatim and analysed using the framework approach.

**Results** This study identified three overarching themes: vulnerability, resilience and support structures. Participants expressed vulnerability relating to the challenging work environment and lack of medications as key stressors resulting in anxiety, stress, anger, isolation and stigmatisation. Signs of resilience with experiences drawn from the 2014 Ebola outbreak, teamwork and a sense of duty were also seen. Peer support was the main support structure with no professional psychosocial support services available to healthcare workers.

**Conclusions** This is the first study to provide evidence of the psychosocial impacts of COVID-19 among front-line healthcare workers in Sierra Leone. Despite signs of resilience and coping mechanisms displayed, they also experienced adverse psychosocial outcomes. There is a need to focus on enhancing strategies such as psychosocial support for healthcare workers and those that overall strengthen the health system to protect healthcare workers, promote resilience and guide recommendations for interventions during future outbreaks.

## INTRODUCTION

Globally, we are experiencing an increasing trend of infectious disease outbreaks which have had a significant impact on livelihoods and public health and safety.[1] Over the last two decades, the 2003 SARS outbreak, the 2009 Influenza (H1N1) pandemic, the 2012 Middle East respiratory syndrome coronavirus

## STRENGTHS AND LIMITATIONS OF THIS STUDY

⇒ To the best of our knowledge, this is the first study to provide qualitative evidence of the psychosocial impacts of COVID-19 on healthcare workers in Sierra Leone.
⇒ Interviewed different cadres of healthcare professionals across different COVID-19 treatment centres which provided a depth of experiences and perspectives on our findings.
⇒ Our study was conducted in 2020 during the first wave of the outbreak, healthcare workers' perceptions and experiences may have evolved over this long pandemic.
⇒ Interviews were conducted remotely, and this may have influenced the degree to which healthcare workers may have responded.

outbreak, the 2013–2016 Ebola virus disease epidemic in West Africa, the 2015 Zika virus disease and the COVID-19 pandemic have all had significant health and non-health impacts.[2 3]

Healthcare workers (HCWs) are a crucial resource in the management of outbreaks and are particularly vulnerable during these periods which brings additional challenges to their already stressful work environment including the increased risk of infection. The pre-existing work stress, coupled with additional outbreak-related stressors combined, may result in psychological distress among the health workforce especially pronounced in front-line HCWs directly involved in the diagnosis, treatment and discharge of patients.[4–7] Therefore, it is essential that policies and strategies that protect front-line HCWs and minimise their psychosocial distress are part of an effective public outbreak response.

In Sierra Leone, during the 2014 Ebola outbreak, HCWs were 30 times more likely to be infected than the public.[8] This resulted in 295 HCW infections and 221 deaths equating

to a 6.85% loss of its total healthcare workforce.[9 10] Poor psychological outcomes including anxiety, post-traumatic stress disorder, anger, fear and community stigmatisation were reported among the few studies conducted during the period which focused on the outbreak's impact on HCWs.[11 12] Post-Ebola outbreak, negative psychological outcomes such as anxiety and anger still lingers among HCWs despite feeling more confident and knowledgeable in providing care for Ebola survivors.[13]

Job-related stressors such as increased workload, lack of personal protection equipment (PPE) and diagnostic capacity, prolonged shift hours, fear of possible infection and death of colleagues were shown to have a profound effect on HCWs' psychological well-being and coping mechanisms during the Ebola outbreak.[12 14 15]

Since the first cases of this novel COVID-19 (SARS-CoV-2) emerged in Wuhan, China and the subsequent declaration of a pandemic on 11 March 2020, over 550 million infections and 6.3 million deaths have been reported as of July 2022.[16] This is inclusive of more than 150 400 COVID-19 infections in HCWs in Africa and 269 in Sierra Leone.[17 18] Furthermore, a recent study in a secondary hospital in Sierra Leone shows a high rate (29%) of secondary COVID-19 infections among HCWs.[19]

Emerging evidence from the COVID-19 pandemic shows HCWs reporting high levels of psychological distress.[20 21] A systematic review and meta-analysis conducted during the first wave of the COVID-19 pandemic supports this evidence with higher levels of anxiety and depression reported among front-line HCWs.[22] However, there is limited evidence from sub-Saharan Africa focusing on the psychological impact of COVID-19 among HCWs in the region.[23]

In Sierra Leone, there are growing concerns that the pandemic may trigger anxiety and post-traumatic stress among HCWs similar to the devastating impact of the 2014 Ebola outbreak.[24] Furthermore, studies conducted to assess COVID-19 preparedness in healthcare facilities in the country have shown HCWs perceive their facilities are ill prepared to respond to the pandemic due to gaps in infrastructure and equipment including inadequate PPE.[25 26]

Given the gaps in the literature, this qualitative study aimed to explore the psychosocial experiences of front-line HCWs during the COVID-19 pandemic in Sierra Leone. The findings of this study could provide evidence for recommendations to guide policy and practice and ensure support structures that build resilience among HCWs within the context of health systems in Sierra Leone are adopted during the COVID-19 pandemic and future outbreaks. Additionally, it will also add to the global, regional and national body of evidence on the psychosocial impact of COVID-19 on HCWs.

## METHODOLOGY
### Study design
An explorative qualitative interview study underpinned by the naturalist research paradigm was conducted. The study drew on literature pertaining to resilience, trust and social support systems to inform our analytical process. Data were analysed using the framework approach, which supports the generation of themes and provided clear systematic steps while maintaining the transparency of the analytic process.[27] The study is based on the Consolidated criteria for Reporting Qualitative research.[28]

### Study setting
This study was conducted in five out of the eight designated COVID-19 treatment centres in Freetown, the capital city of Sierra Leone and the epicentre of the COVID-19 outbreak.[29] During the 2014 Ebola outbreak in Sierra Leone, Freetown was also one of the epicentres of the outbreak and recorded the highest prevalence (39%) of confirmed cases across the country.[30]

In Sierra Leone, the management of COVID-19 patients was separated from regular health service provision to protect and maintain essential health services based on learnings from the 2014 Ebola outbreak. A total of 21 treatment centres were designated for COVID-19 across the country.[31] These treatment centres were classified into three categories based on the severity of the disease. Isolation units (suspected cases), community care centres (asymptomatic and mild cases) and COVID-19 treatment centres (moderate, severe and critical cases, and all patients over 65 years with underlying medical conditions regardless of the severity of their symptoms).[31]

### Sample and recruitment
We recruited thirteen front-line HCWs to complete in-depth interviews across five COVID-19 designated treatment centres in Freetown. Participants were interviewed in July and August 2020. Sampling was purposive to include a range of different cadres, employment statuses, gender, work experience and professional roles (table 1). All participants also had prior experience providing care during the 2014 Ebola outbreak. These participants were

**Table 1** Characteristics of front-line healthcare workers who participated in the study, July–August in Freetown, Sierra Leone, 2020

| Participants | 13 |
| --- | --- |
| Cadres | Doctors (6) Nurses (6) Community Health Officer (1) |
| Gender | Male (7) Female (6) |
| Employment status | Government (7) Military (3) Volunteers (3) |
| Work experience | 5–10 years (11) >10 years (2) |
| Location (work site) | COVID-19 treatment centres (6) Community care centres (3) Isolation units (2) National COVID-19 Emergency Response Centre (3) |

recruited via telephone calls, emails and social media networking platforms, using our pre-established communications and connections. Two of the researchers were members of the national COVID-19 response which made it easy to identify study participants and share study information with consented participants.

## Data collection

We conducted semistructured, one-to-one interviews exploring the psychological and social impacts of the COVID-19 pandemic on participants. The interviews were conducted by SMT (lead researcher) remotely via Zoom or WhatsApp platforms using a pretested topic guide. The development of the topic guide was guided by existing behavioural change theories on health stress and coping mechanisms.[32] This pretested topic guide has been provided as online supplemental material. This ensured we collected the information within a short time, minimising disruption to the busy schedules of participants.[33 34] It was also appropriate given the COVID-19 social distancing restrictions which were in place.[35 36]

The interviews lasted for about 45–60 min and were done in Krio (local dialect) and English (depending on participant preference) using a topic guide. All participants were given a participant information sheet a week before the interviews and encouraged to ask questions. Written informed consent was obtained prior to interviews and a demographics form was completed by all participants. We audiorecorded interviews with participants' consent, and recordings were transcribed by the lead researcher (SMT) who has experience in qualitative research. Interviews conducted in Krio were listened to and transcribed directly in English by the lead researcher (ST) who is fluent in both languages. Where some words or concepts were hard to understand or directly translate a third opinion from other researchers (IFK, DBA and FFK) was sought as well as respondent validation. The audiorecording of interviews, interview notes and available transcripts enabled repeated revisiting of data to remain immersed in the data and to participants' original accounts, helping to enhance the validity of the results.

As this was a rapid study to provide evidence to support the ongoing COVID-19 response, interviews ceased when data saturation was reached as a commonality between themes emerged and the lead author identified no new themes emerging within the interviews.

## Patient and public involvement

The study participants or the public were not involved in the design and conduct of the study. However, preliminary findings and recommendations have been shared with the case management pillar of the National COVID-19 Emergency Response centre and the Ministry of Health and Sanitation. Recommendations are designed to support improving the development of policy and practice to ensure HCWs are provided with psychosocial support during the pandemic and future disease outbreaks. The findings will be shared with the wider public and on social media.

## Data analysis

Following anonymisation by the lead researcher (SMT), transcripts were uploaded to NVivo V.12 to aid analysis. Using the framework analysis approach, three researchers (SMT, ST and LD) initially read transcripts independently and discussed any emerging codes of potential significance to the research objectives. A coding framework was developed both deductively from the topic guide and study objectives, and inductively from themes emerging from the data. This coding framework was applied to each transcript with charts developed for each theme. All study authors agreed on the final themes and included rich and verbatim descriptions of participants' experiences to support these findings.

## Study trustworthiness

Respondent validation was used during the interview process to clarify responses; feeding back the information provided to participants ensured their responses were accurately captured and understood. This feedback process was conducted in more detail with interviews conducted in the local dialect to avoid misinterpretation even though the lead researcher is fluent in both languages to improve the credibility and confirmability of findings. Triangulation of findings from the data collection was also checked against secondary data from the literature to further aid credibility and transferability.

The positionality of the lead researcher (SMT) as a Sierra Leonean medical doctor and public health professional and with two other researchers (IFK and SS) being part of the COVID-19 response, ensured the establishment of rapport and mutual trust to aid data collection and to understand the local context and healthcare system. The methodological approach and findings were shared with key stakeholders to support dependability and our results include thick verbatim descriptions and quotes from participants to support our findings.

## RESULTS

The result is structured into three key interlinking themes, such as vulnerability, resilience and support systems, each with several subthemes (table 2).

### Vulnerability

#### Work environment

Despite acknowledging that HCWs are used to working under stressful conditions, increased experiences of stress, anxiety and isolation were expressed during COVID-19 patient care. These experiences peaked during the first 3 months of the COVID-19 pandemic and seemed to be more vividly expressed by females and nurses than by males and doctors.

> Working under stressful conditions is normal in Sierra Leone… (HCW 08, Doctor, Male)

**Table 2** Themes and subthemes of the psychosocial impacts of COVID-19 on front-line healthcare workers in Freetown, Sierra Leone, 2020

| Theme | Subtheme |
| --- | --- |
| 1. Vulnerability | 1.1. Work environment<br>1.2. Lack of resources and medication<br>1.3. Stigmatisation<br>1.4. Post traumatic stress disorder<br>1.5. Lack of recognition |
| 2. Resilience | 2.1. Essential healthcare service provision<br>2.2. Teamwork<br>2.3. Training<br>2.4. Sense of purpose/duty<br>2.5. 2014 Ebola outbreak |
| 3. Support Systems | 3.1. Peer support<br>3.2. Donations<br>3.3. Professional support<br>3.4. Challenges in the health system |

When the first cases came, we were not enough at all…that time it was hell, because we had 30 beds and 4 nurses cannot deal with 30 patients… (HCW 01, Nurse, Female)

HCWs feeling overwhelmed were predominantly associated with disruptions in the work environment with increased workload, inadequate human resources and a rapid influx of COVID-19 patients especially the critically ill.

The fear of possible infection due to patient care also fed into HCWs' vulnerability with concerns for personal safety and onward transmission to family members and friends.

When a colleague became infected, it was really stressful…we did not know where she got the contact…we were quarantined, samples collected…we were scared…what if our samples turn out positive? what will we do? …we have put our lives and families at high risk… (HCW 06, Nurse, Female).

### Lack of resources and medications

Doctors and in-charge nurses with more managerial roles expressed anger, guilt, helplessness and frustration related to the medical commodities such as medications, oxygen cylinders and personal protective equipment at COVID-19 treatment centres.

The first death in the centre was diabetic and insulin was unavailable in the unit… it is challenging when you do not have what you need at the right time…. knowing at the back of your mind that if something goes wrong, you are to blame so it was very stressful. (HCW 08, Doctor, Male).

We faced constraints with oxygen supply and if you have three patients in need it is difficult to know how to share it…I was monitoring a patient when I saw the oxygen cylinder was almost empty so I called and was told there was no gas…I was still on the call when the patient died… I just stood there…I felt guilty that there was something that could have been done…so it affected me for a few days…it ate at my conscience… (HCW 01, Nurse, Female)

### Stigmatisation

Stigmatisation either from colleagues not involved in the response and from the community resulted in HCWs feeling isolated and depressed. These feelings were equally expressed across all cadres and gender.

Our colleagues were scared for us, knowing that being at the frontline, we were at risk of becoming infected…in fact, they were also afraid of us, anytime we went to the regular hospital setting, they told us not to come close to them… (HCW 02, Nurse, Female)

When COVID-19 started, my family told me not to come home…they said I should not have joined the response…so I packed my things and went to stay in the hospital. I was in the hospital for over 1 month before I came home. (HCW 05, Nurse, Male)

### Post-traumatic stress disorder symptoms

These findings also highlighted that the negative experiences from the 2014 Ebola outbreak such as fear and anxiety were triggered and led to some of the psychosocial outcomes front-line HCWs faced during the COVID-19 response. This was reflected in the inappropriate use of PPE despite limited supplies.

The Ebola experience brought up some PTSD… when this corona started we observed that health workers were using too many PPE…because that was what they did during the Ebola and the fear remembering that colleagues died from Ebola… (HCW 11, Doctor, Male)

### Lack of recognition

Front-line HCWs also reported feeling that their hard work was not adequately recognised, and appreciated by the government, especially the Ministry of Health and Sanitation. The lack of visitation by top government officials to COVID-19 treatment centres was mentioned by four HCWs as particularly discouraging.

When the good comes, you hardly hear about the nurses… instead it is the big guns that are boasting…. what about the health workers that are in the fight… It's like nobody even cares…the minister has not even come around…those little things count. (HCW 10, Nurse, Female)

A certificate of recognition should also be given to health workers in the frontline…this will help show

that our efforts are valued and appreciated… (HCW 13, Doctor, Male)

## Resilience

Despite the challenges, pressures and vulnerabilities, front-line HCWs also demonstrated resilience, growth, pride and job satisfaction during this period.

### Essential healthcare services provision

The COVID-19 pandemic caused a significant shock to the health system and despite the challenges in the work environment and shortage of medications and commodities, medical doctors mainly viewed the COVID-19 response as a success as health services adapted, were maintained and did not collapse.

> In terms of successes, the health system has not collapsed…. We quickly move from 30-bed, decentralised, and have treatment facilities in almost every district…. I think we have 820 beds…. our surveillance is also good at pursuing contact… (HCW 09, Doctor, Male)

> Sometimes we get shortage of PPE, but they fast track the process, so we have been able to maintain adequate PPE supplies, we have not reached the stage to recycle PPE…this has helped us to feel safe… (HCW 04, Doctor, Male)

### Teamwork

Working long, socially isolating and tedious hours during the early months of the pandemic resulted in a close-knit group of HCWs across the different COVID-19 treatment centres. The resulting sense of camaraderie and teamwork was seen as a key factor for HCWs to continue to provide healthcare services.

> We watch each other's back and help each other with tasks…this is one of the biggest successes we have seen during the period. (HCW 04, Doctor, Male)

Inexperienced HCWs, especially those recruited from other wards, were most appreciative of the close collaboration with senior colleagues as this improved their confidence.

> When I joined the treatment centre from another unit, I found it difficult to adapt… we had a lot of patients, and it was not easy…but our senior colleagues taught and supported us …. this really help me, and I became less afraid… (HCW 03, Nurse, Female)

The HCWs also viewed collaborative and technical skills gained during COVID-19 patient care as making them more equipped and better prepared for future work.

> It is challenging but also a learning process as I have to deal with people from different sectors and backgrounds, not just the ones normally involved in the healthcare system… (HCW 12, Doctor, Male)

> I have gained a lot of experience because I did not know how to use oxygen concentrators… but I can… (HCW 03, Nurse, Female)

### Training

HCWs reported training centred around infection prevention and control (IPC) including the use of PPE and case management was the primary form of preparation for COVID-19 patient care. Training sessions were organised before the country's first COVID-19 case was reported on 31 March 2020 and this reduced anxiety over possible personal safety and HCW infections. These training(s) also built on experiences gained during the 2014 Ebola outbreak which strengthened HCWs' competencies in IPC.

> We did refreshers trainings on how to put on PPE, IPC and hand hygiene before the COVID outbreak came into the country….so when COVID came we were not surprised… (HCW 12, Doctor, Male)

> I think the Ebola gave us a footing especially with IPC…so it was not new…I think this is less stressful compared to the Ebola Era. (HCW 07, CHO, Female).

### Sense of duty

All participants reported reminding themselves of their Hippocratic oath and the nature of their duties as a significant way of dealing with stress. Empathy towards the COVID-19 patients was also a strong motivator to continue patient care during the stressful period, with HCWs feeling unable to stay away and stop working despite the challenges and psychosocial outcomes.

> When I feel weak and don't want to go work I remember my nurse's and military pledge so I need to fulfil my duty…I can't stay home because you feel that it's your mom or daddy that is in the hospital bed, you need to go help… (HCW 03, Nurse, Female)

### 2014 Ebola outbreak

All the HCWs expressed that experiences gained while working during the 2014 Ebola outbreak were invaluable as they were more confident, better prepared psychologically and had transferable skills and competencies which reduced their levels of fear for providing care during the COVID-19 pandemic. Most participants reported feeling more afraid and stressed during the Ebola outbreak as compared with the COVID-19 pandemic demonstrating significant resilience, despite the presence of emotional triggers as previously described.

> The emotions during the Ebola was scarier than the COVID because that was the first outbreak exposure, COVID is my second so I did not have much fear to join the response because I had previous experience… (HCW 13, Doctor, Male)

> The experience I got during Ebola was greater than this because during Ebola people were dying almost every minute…you needed great courage to work…

we wore diapers inside the PPE, because if you left the ward you would not want to come back in… (HCW 05, Nurse, Male)

## Support structure

Front-line HCWs expressed vulnerability and resilience were influenced by their support structures.

### Peer support

Gestures such as sharing meals, discussions before and after shifts as well as sharing jokes were expressed as the most common form of support. Several nurses expressed that encouraging words from doctors in charge of the COVID-19 unit also served as a boost. A WhatsApp group was used by some of the front-line HCWs to express their feelings, seek guidance and foster bonding among peers.

We have this forum that everybody can vent out through audio clips, the experience of the day so we support each other and give advice on how to cope with the stress… (HCW 10, Nurse, Female)

When I feel weak…I listen to gospel songs…when I do this I feel Ok within myself…God gives me the energy and strength to be able to cope with the work. (HCW 06, Nurse, Female)

### Donations

Donations from organisations and the community to COVID-19 treatment centres came in as medications, equipment and personal items for COVID-19 patients, with a few targeting the HCWs. These gestures were received as a welcoming boost improving both patient care and morale among the staff.

The day we received donations and had enough medications and some of the equipment we needed in the treatment centre…it lifted the mood of the entire staff, people were willing to work, absenteeism reduced…that was a big game changer. (HCW 12, Doctor, Male)

### Professional support

Although there were psychosocial support services for patients in the treatment centres, this service is not accessible to front-line HCWs involved in the treatment of admitted COVID-19 patients. This was met with frustration from being excluded from such services by some HCWs.

We have gone through the Ebola, mudslide, flooding now COVID, I know Sierra Leonean we are resilient, but we should not take it for granted…we are traumatised and full of anxiety and nobody cares about our emotional stress… (HCW 10, Nurse, Female)

### Health system challenges

Some participants expressed a sense of missed opportunity to capitalise on the country's history of outbreaks to strengthen its health system.

During Ebola we had support from all over the world but with COVID-19, everyone has been grounded in their respective countries…this is a way of telling us to strengthen our health sector… (HCW 13, Doctor, Male)

Improving conditions of service, availability of essential medications and supplies, the recruitment of volunteer HCWs not on the government payroll and the provision of health insurance for HCWs were repeatedly expressed.

I think if we have the psychosocial counselling structure backed up by hospitals having the basics…this would help healthcare workers during outbreaks… (HCW,11, Doctor, Male)

## DISCUSSION

This qualitative study explored the psychosocial experiences of front-line HCWs delivering COVID-19 care in Sierra Leone. The findings presented in this study are a particularly valuable in-depth qualitative analysis of psychosocial distress and extreme resilience among HCWs in the COVID-19 response. To date, the experiences of front-line HCWs who have been involved in both the 2014 West Africa Ebola outbreak and the COVID-19 pandemic are largely absent within the debate on psychosocial distress during epidemics. We present our conceptual framework (figure 1), to show that front-line HCWs' psychosocial outcomes in Sierra Leone during the COVID-19 outbreak, are shaped by their vulnerabilities and resilience, with these experiences inexplicably linked to the strengths and weaknesses of the health system.

Similar to other COVID-19 studies, our study also shows that stress, anxiety, fatigue, isolation and fear of infection affected front-line HCWs the most during the start of the pandemic.[37 38] This is not surprising given the novelty of the disease and the traumatic nature of infectious disease outbreaks[39] We would, therefore, recommend that psychosocial support services be implemented early to promote resilience and protect HCWs.

Furthermore, studies done in sub-Saharan Africa, Asia and Europe during the COVID-19 pandemic emphasised similar themes to those we identified that the lack of medications, medical equipment and oxygen supply introduced further constraints in the workplace, affecting HCWs' ability to carry out their duties, frequently resulting in psychological distress.[20 40 41] The shortage of PPE was highlighted in systematic reviews as a major predisposing factor for negative psychological impact on HCWs during the COVID-19 pandemic.[7 42] However, our findings contradict this trend as HCWs did not express significant concerns in terms of the availability of PPE and were proud that the healthcare systems did not collapse. Despite this, the importance of PPE to HCWs in managing their fears and vulnerabilities was apparent and working with HCWs to achieve appropriate use of PPE based on disease conditions would support in ensuring

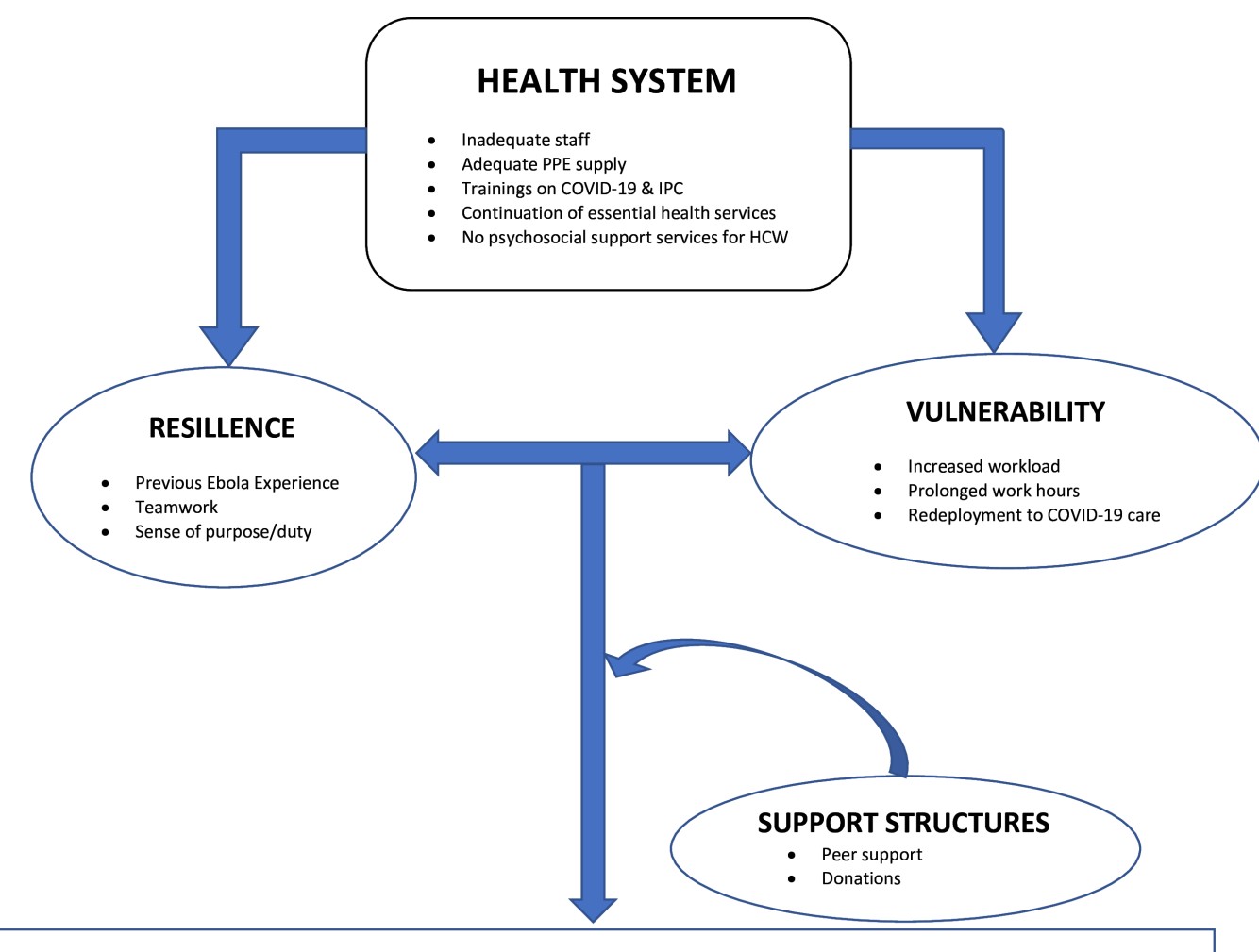

Figure 1 Conceptual framework showing the interlinkages that shape the psychosocial impact of COVID-19 on front-line healthcare workers in Sierra Leone, 2020. HCW, healthcare worker; IPC, infection prevention and control; PPE, personal protection equipment.

efficient use of scarce health system resources. We recommend that investments in health system strengthening efforts should be prioritised as this is crucial in outbreak response to ensure the effectiveness of HCWs, improve safety and reduce stress.[36]

Although all participants expressed psychological experiences, there were variations in the perceived levels based on gender, cadre and work experience. This study confirms findings from Ghana and the UK, which highlighted that nurses are more vulnerable given their contact time with patients and in China which states that inexperienced HCWs express more stress while trying to adapt to the new settings.[43–45] Further explanation for the gender-related differences in our study may be because most of the female participants are nurses and males are doctors. Reflecting on this finding and considering gendered occupational segregation in the health workforce in Sierra Leone, with female nurses accounting for 90% of all nursing personnel;[46] gender considerations must be prioritised when planning psychosocial support

services for HCWs, together with regular training and simulation exercises in emergency response to allay fear and anxiety among inexperienced HCWs during future outbreaks.[21 29 38]

Our study adds to the overall body of evidence emerging from other studies on HCWs during outbreaks that direct involvement with patients had a varying effect on HCWs' relationships often resulting in stigmatisation increasing their psychological distress.[3 12 37] This signifies the need for the early introduction of risk communication and community engagement programmes during outbreaks to increase awareness and reduce stigma.

Resilience among HCWs during the COVID-19 pandemic is well documented in the literature.[40 47] In our study, however, signs of resilience were mainly because of the legacy of the 2014 Ebola outbreak, a finding which is likely transferable to a limited number of contexts. The experience gained from patient care during the Ebola outbreak was seen as training in outbreak response and formed part of HCWs' coping strategy. Our finding

further showed that HCWs expressed optimism that the foundations laid by the Ebola outbreak have been built on by the COVID-19 pandemic. Hence, they possess the necessary skills and expertise to face future outbreaks. This growing theme of resilience is also shown in the findings of a recent study to assess the impact of the COVID-19 pandemic on hospital utilisation in Sierra Leone.[36] It is worth acknowledging that despite the different nature and trajectory of both outbreaks with varying responses, there are common issues that inevitably shape the psychosocial outcomes of HCWs.

Measures such as training and availability of IPC measures have been found to have a pivotal impact in reducing front-line HCWs' risk of infections and a positive effect on their psychosocial outlook.[3 4] Findings from our study resonate with this body of evidence as multiple trainings were conducted for HCWs and the management of COVID-19 patients was separated from general care. As a result, participants felt safer, less anxious and more confident to provide patient care.[31] Moving forward, training a critical mass of HCWs that possess the prerequisite knowledge, skills and competencies to provide quality healthcare to infectious disease patients that are admitted to a designated treatment centre could prove invaluable.

In line with previous studies done to assess motivators that kept HCWs working during outbreaks, we showed that the pandemic presented an opportunity for inexperienced front-line healthcare to learn new skills and develop stronger teams. Professional ethics, empathy, religion and community donations helped HCWs in coping with emotional struggles and physical exhaustion and focus on performing their duty.[39–41]

Broader literature on support structures used by HCWs during outbreaks showed that HCWs discussed challenges and stresses with colleagues at the end of the shift and on social media platforms.[4] Our findings are in concordance with this peer support theme. However, the unavailability of counselling and professional support services for HCWs in our study resonates with findings from studies done in South Africa and Ghana but contrasts with those done in Europe where formal psychosocial support services were provided for HCWs.[20 38 44] This is disconcerting even with signs of resilience and success. It signifies a missed opportunity and gap presenting further evidence for the need to instil measures such as counselling services, the establishment of help hotlines and friendly relaxation spaces[48] to acknowledge and protect HCWs from undue exposure and adverse psychosocial outcomes during crises.

Psychosocial support structures must also go together with activities that strengthen the healthcare facilities given that the work environment is a strong predisposing factor for psychosocial distress. Furthermore, improvement of the conditions of service for HCWs with the introduction of health insurance schemes, employment of volunteering HCWs and formal recognition of service and appraisal systems for front-line HCWs with exemplary performance during outbreak response would boost morale as well as reinforce positive conduct.

## Strengths and limitations

To the best of our knowledge, this is the first study to provide qualitative evidence of the psychosocial impacts of COVID-19 on HCWs in Sierra Leone. We interviewed different cadres of healthcare professionals across five designated COVID-19 treatment centres which provided a depth of experiences and perspectives on our findings.

There were some limitations. Our study was conducted in 2020 during the first wave of the outbreak, HCWs' perceptions and experiences may have evolved over this long pandemic. However, this study captured the experiences of HCWs at the height of the pandemic. Furthermore, recently published studies have highlighted similar experiences of HCWs that are involved in COVID-19 patient care.[32 49]

The interviews were conducted remotely, and this may have influenced the degree to which HCWs may have responded. However, prior established rapport and communications with participants on social media and through emails before the interview process helped to circumvent this potential challenge. This is in keeping with recommendations from other qualitative studies that used remote interviewing.[50 51] This also allowed us to conduct interviews at the participant's preferred time, minimised distraction and was appropriate given the strict COVID-19 restrictions in place.

## CONCLUSION

Our study is the first to explore the psychosocial impacts of COVID-19 among front-line HCWs in Sierra Leone. These different cadres of front-line HCWs experienced a wide range of psychosocial outcomes including anxiety, stress, guilt, anger, empathy, stigmatisation and a sense of duty. Strong signs of resilience were also expressed due to the legacy of the 2014 Ebola outbreak. We, therefore, recommend that strategies such as formal psychosocial support for HCWs, improved healthcare supply chain and the training of a critical mass of HCWs on outbreak preparedness and response be developed and implemented urgently to promote resilience and strengthen the health system in preparation for future epidemics.

**Acknowledgements** The authors would like to thank those who gave their time to participate and contribute to the study.

**Contributors** SMT conceived the initial study and IFK, ST and LD contributed to study design and ethical approval process. SMT was responsible for data collection and was assisted by IFK, DBA, FFK and SS during this stage, SMT conducted data analysis alongside ST and LD. SMT produced the original draft of the manuscript. All authors critically reviewed and edited and provided comments on the first and subsequent drafts. All authors approved the final manuscript for submission. SMT is the guarantor.

**Funding** This work was supported by the UK Research and Innovation (UKRI). The GCRF Accountability for Informal Urban Equity Hub ('ARISE') is a UKRI Collective Fund award with award reference ES/S00811X/1. The write up was also supported by a Ken Newell Bursary award.

**Competing interests** None declared.

**Patient and public involvement** Patients and/or the public were involved in the design, or conduct, or reporting, or dissemination plans of this research. Refer to the Methodology section for further details.

**Patient consent for publication** Not applicable.

**Ethics approval** The study was reviewed and approved by the Sierra Leone Ethics and Scientific Review Committee (24 July 2020) and Liverpool School of Tropical Medicine Masters Review Panel (20(02)a). Written informed consent was obtained from all participants.

**ORCID iDs**
Sia Morenike Tengbe http://orcid.org/0000-0001-7287-4426
Ibrahim Franklyn Kamara http://orcid.org/0000-0003-1454-4650
Desta B Ali http://orcid.org/0000-0002-7687-8341

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
