## [Reviewer comments · BMJ Open]

ARTICLE DETAILS

TITLE (PROVISIONAL)	The Psychosocial Impact of COVID-19 Pandemic on Frontline Healthcare Workers in Sierra Leone: an explorative qualitative study
AUTHORS	Tengbe, Sia Morenike; Kamara, Ibrahim Franklyn; Ali, Desta B; Koroma, Fanny F; Sevalie, Stephen; Dean, Laura; Theobald, Sally

VERSION 1 – REVIEW

REVIEWER	Umeokonkwo, Chukwuma Alex Ekwueme Federal University Teaching Hospital Abakaliki, Department of Community Medicine
REVIEW RETURNED	23-Jan-2023

GENERAL COMMENTS	Thank you for the opportunity to review this research work that qualitatively explored the psychological experiences of the frontline health workers in Sierra Leone. The work is well written and is topical. The authors are encouraged to consider the following suggested edits to help improve the understanding of the work. Method Study setting: How many treatment centres were available in the capital city (Freetown) to enable the reader have a sense the selection. Interview guide/ Topic guide The authors need to describe the process of the development or adoption/adaptation of the interview guide and its pretesting. The authors should consider providing a copy of the interview guide as a supplementary material in the manuscript. Line 152: The authors need to write out abbreviations and acronyms in full the first time they are used (SMT) Results Line 209: It is not clear what "These experiences..." was referring to having quoted nurse HW01, the authors need to clarify whether they were referring to the anger, anxiety and isolation or the hellish work environment the HW was referring to. The authors should consider rephrasing the sentence for clarity. Line 242: "These resulting feelings of..." The authors should consider rephrasing this sentence to show that stigmatisation from health workers and community members resulted in the feeling of isolation and depression among the frontline health workers. The write up under the post-traumatic stress disorder symptom could be moved to the work environment sub theme or lack of resources and medication sub theme. Since the theme do not
--

	focus on the types of the psychological symptoms expressed by the frontline health workers. Line 389: Lack of recognition appear to me more like vulnerability and the content appear best suited to the work environment vulnerability. The content could be move to that section. Discussion The authors need to also draw comparison to similar relevant studies carried out in West Africa. Line 454-465: The study was a qualitative study with 13 participants, the design and population was not appropriate to draw conclusions on gender differences in the psychological experiences. The authors should consider rephrasing this section to avoid such comparisons which were neither part of the objectives nor drawn from the result. Line 525-526: The authors should consider removing wide from the sentence as there are many cadre of health workers involved in the pandemic who were not involved in the study. Line 581: The authors should specify the type of informed consent obtained from the participants
--	---

REVIEWER	James, Peter University of Sierra Leone College of Medicine and Allied Health Sciences, Pharmaceutical Sciences I have published with one of the co-authors. However, that did not influence my review.
REVIEW RETURNED	08-Feb-2023

GENERAL COMMENTS	Introduction, 1. Third paragraph page 4 page lines 85-88: What about the post-outbreak impact such as providing care to survivors of the Ebola outbreak? You may want to consider this article James PB, Wardle J, Steel A, Adams J, Bah AJ, Sevalie S. Providing healthcare to Ebola survivors: A qualitative exploratory investigation of healthcare providers' views and experiences in Sierra Leone. Global Public Health. 2020 Sep 1;15(9):1380-95. 2. Fourth paragraph page 4 page lines 90-93: Such anxiety maybe related to the fact they considered their health facilities as ill prepared to adequately respond to COVID. Infact majority mentioned that their facilities lack adequate PPE. You may want to consider the following article Kanu S, James PB, Bah AJ, Kabba JA, Kamara MS, Williams CEE, Kanu JS. Healthcare Workers' Knowledge, Attitude, Practice and Perceived Health Facility Preparedness Regarding COVID-19 in Sierra Leone. J Multidiscip Healthc. 2021 Jan 11;14:67-80. doi: 10.2147/JMDH.S287156. PMID: 33469299; PMCID: PMC7810694 METHODS 1. The authors need to clearly indicate the study design. Yes, it is qualitative but is deductive or inductive? secondly, what theoretical framework underpinned the study design? if no theoretical framework was used the authors need to proffer the reason(s) why. 2. Please indicate who transcribed the audio recordings and what is/are their competency(ies) if any to do such a task 3. page 9 page lines 162-164: The authors need to properly describe the trustworthiness of the study findings by describing how credibility, transferability, confirmability, and dependability were ensured.
---

	4. Data analysis: The description of the data analysis look scanty and vague. For example, readers would like to know who did the data analysis and how many, and what competency this person or people have to analyze the data. Thirdly, the authors mentioned the framework approach was used for data analysis, and as they mentioned it has clear systematic steps. However, they failed to fully describe these steps. I suggest that the authors fully describe how they follow these steps Results Page line 199: I suggest the quote come after the summary describing the subtheme Discussion 1. There are sentences throughout the discussion that need to be referenced. examples Pages lines 435-436, 451-452, 489-491 and 535-536 2. In page lines 509-512, you stated the need to instill measures to acknowledge and protect HCWs. Can the authors mention examples of some of these measures?
--	---

REVIEWER	Gupta, Snehil AIIMS Bhopal, Department of Psychiatry
REVIEW RETURNED	05-Mar-2023

GENERAL COMMENTS	I congratulate authors for conducting a valuable research work and nicely drafting the manuscript. I thoroughly enjoyed reviewing it. There are few observation which if authors address would add rigour to the methodology and comprehensiveness in the paper. They are as follows: 1) Line 110: The findings of this study will.....The statement cannot be definite. Better to represent a possibility (could/can). 2) 114: Aim-Better would be to also highlight qualitative nature of work here. 3) Data analysis: Authors should provide more information about data interpretation, e.g., triangulation-was it performed or not, Who performed coding and was it rechecked by other investigators or not? Likewise, the categorization of the themes were done by whom? how the consensus was built? 4) Word count should be restricted to 4000, as per the Journals' guideline. Discussion may be trimmed.
---

VERSION 1 – AUTHOR RESPONSE

Reviewer 1

Overall comment of Reviewer 1

Thank you for the opportunity to review this research work that qualitatively explored the psychological experiences of frontline health workers in Sierra Leone. The work is well-written and topical. The authors are encouraged to consider the following suggested edits to help improve the understanding of the work.

Response to Reviewer 1 overall comment

Thank you very much for taking the time to review our article. We will act on all the input, comments, and suggestions accordingly.

Comment 1

Method

Study setting: How many treatment centres were available in the capital city (Freetown) to enable the reader to have a sense of the selection.

Response 1

Thank you very much for your comment. There were eight (8) COVID-19 treatment centres in the capital city (Freetown). This has been included in the manuscript in line 139 of the revised version.

Comment 2

Interview guide/ Topic guide

The authors need to describe the process of the development or adoption/adaptation of the interview guide and its pretesting. The authors should consider providing a copy of the interview guide as supplementary material in the manuscript.

Line 152: The authors need to write out abbreviations and acronyms in full the first time they are used (SMT)

Response 2

Thank you for the comments. The process of developing, and adaptation of the topic guide has been included in the manuscript in lines 173- 175 of the revised version. The topic guide has also been added as supplementary material to the main manuscript. Additionally, SMT is the initial for the lead researcher and this clarification has been made in the manuscript in line 173 of the revised version.

Comment 3

Results

Line 209: It is not clear what "These experiences..." was referring to having quoted nurse HW01, the authors need to clarify whether they were referring to the anger, anxiety and isolation or the hellish work environment the HW was referring to. The authors should consider rephrasing the sentence for clarity.

Line 242: "These resulting feelings of..." The authors should consider rephrasing this sentence to show that stigmatisation from health workers and community members resulted in feeling of isolation and depression among frontline health workers.

Response 3

Thank you for the comments. The sentence concerning the experiences of HW01 has been paraphrased to provide the reflection of healthcare workers feeling overwhelmed by the conditions of the work environment. Please see line 239 of the revised manuscript with the clear experiences of healthcare workers. This provides more clarity.

The sentence concerning the stigmatization of healthcare workers has been rephrased for clarity and to reflect the suggested changes. Please see lines 272-274 of the revised manuscript

Comment 4

The write-up under the post-traumatic stress disorder symptom could be moved to the work environment sub-theme or lack of resources and medication sub-theme. Since the theme do not focus on the types of psychological symptoms expressed by frontline health workers.

Line 389: Lack of recognition appears to me more like vulnerability and the content appears best suited to the work environment vulnerability. The content could be moved to that section.

Response 4

Thank you for your comment, post-traumatic stress disorder is a sub-theme under the vulnerability theme. Additional information has been provided to justify it being a stand-alone subtheme (line 286-288). We believe that past experiences of the Ebola outbreak play a key role in the vulnerability felt by HCWs and was related to some of the fear and anxiety experienced by healthcare workers resulting in the irrational use of PPEs despite limited supplies. Additionally, the suggestion for the rearrangement of the lack of recognition has been made as it has been moved to the vulnerability sub-theme in Table 2 (line 222) and the text moved to the write-up sections (Lines 295-308).

Comment 5

Discussion

The authors need to also draw comparisons to similar relevant studies carried out in West Africa.

Line 454-465: The study was a qualitative study with 13 participants, the design and population were not appropriate to draw conclusions on gender differences in the psychological experiences. The authors should consider rephrasing this section to avoid such comparisons which were neither part of the objectives nor drawn from the result.

Line 525-526: The authors should consider removing wide from the sentence as there are many cadres of health workers involved in the pandemic who were not involved in the study.

Line 581: The authors should specify the type of informed consent obtained from the participants.

Response 5

Thank you for the comments. The majority of studies conducted on this topic were done outside of Africa, however, the few published studies from Africa were included in the revised manuscript. We included studies conducted in Ghana, (line 497) and South Africa (line 554) to the manuscript which were the published studies available during the write-up of the initial manuscript. Furthermore, the gender difference was obtained from our study findings. An example can be seen in lines 230-234 where females experienced more stress, anxiety and isolation than males. We believe that this should stand out to suggest further research in this area.

The sentence where a wide range of healthcare workers was used has been rephrased, please see line 574 of the revised manuscript. We have specified the type of informed consent obtained from the participants. Please see line 634 of the revised manuscript.

Reviewer: 2

Comment 1

Introduction,

Third paragraph page 4 page lines 85-88: What about the post-outbreak impact such as providing care to survivors of the Ebola outbreak? You may want to consider this article

James PB, Wardle J, Steel A, Adams J, Bah AJ, Sevalie S. Providing healthcare to Ebola survivors: A qualitative exploratory investigation of healthcare providers' views and experiences in Sierra Leone. *Global Public Health*. 2020 Sep 1;15(9):1380-95.

Response 1

Thank you for the comment. The post-outbreak impact of providing care to Ebola survivors has been included in the revised manuscript in lines 96-98.

Comment 2

Fourth paragraph page 4 page lines 90-93: Such anxiety may be related to the fact they considered their health facilities as ill-prepared to adequately respond to COVID. In fact, the majority mentioned that their facilities lack adequate PPE. You may want to consider the following article Kanu S, James PB, Bah AJ, Kabba JA, Kamara MS, Williams CEE, Kanu JS. Healthcare Workers' Knowledge, Attitude, Practice and Perceived Health Facility Preparedness Regarding COVID-19 in Sierra Leone. *J Multidiscip Healthc*. 2021 Jan 11;14:67-80. doi: 10.2147/JMDH.S287156. PMID: 33469299; PMCID: PMC7810694.

Response 2

Thank you for the comment, the fourth paragraph in reference refers to job-related stressors healthcare workers experienced during the Ebola outbreak and not the COVID-19 outbreak. We believe it is important for us to mention such in the manuscript. Furthermore, thank you very much for sharing the COVID-19 article with us, it makes a good read.

Comment 3

Method

The authors need to clearly indicate the study design. Yes, it is qualitative but is deductive or inductive? secondly, what theoretical framework underpinned the study design? if no theoretical framework was used the authors need to proffer the reason(s) why.

Response 3

Thank you for your comment. The study design and the framework used have been included in the revised manuscript. Please see lines 132-136 of the revised manuscript.

Comment 4

Please indicate who transcribed the audio recordings and what is/are their competency(ies) if any to do such a task

Response 4

Thank you very much for your comment. We have included the person that did the transcribing with their competencies. Please see line 185- 186 for the revised manuscript. The lead researcher conducted the transcribing and has experience in qualitative research.

Comment 5

page 9 page lines 162-164: The authors need to properly describe the trustworthiness of the study findings by describing how credibility, transferability, confirmability, and dependability were ensured.

Response 5

Thank you for the comment, the authors described the steps taken to ensure trustworthiness in the data analysis section in lines 205-214 and in the strength and limitation section in lines 576-578; 587-594 of the revised manuscript. This was not included in lines 162-164 of the original submission as it has been explained clearly in the data analysis and strengths and limitations sections.

Comment 6

Data analysis: The description of the data analysis looks scanty and vague. For example, readers would like to know who did the data analysis and how many, and what competency this person or people have to analyze the data. Thirdly, the authors mentioned the framework approach was used for data analysis, and as they mentioned it has clear systematic steps. However, they failed to fully describe these steps. I suggest that the authors fully describe how they follow these steps

Response 6

Data analysis was led by the lead author and other co-authors. The data analysis section has been updated to provide specific details on the author's participation in the different stages of the data analysis process. Please see Lines 205-214 of the revised manuscript.

Comment 7

Results

Page line 199: I suggest the quote come after the summary describing the subtheme.

Response 7

The authors believe the introduction of the quote before the summary sets the scene for the proceedings paragraph. A similar approach is used in published qualitative studies.

Comment 8

Discussion

There are sentences throughout the discussion that needs to be referenced. examples Pages lines 435-436, 451-452, 489-491, and 535-536

Response 8

Thank you very much for your comment. We have included all the relevant in-text citations in the discussion section and generated reference lists as suggested.

Comment 9

On page lines 509-512, you stated the need to instill measures to acknowledge and protect HCWs. Can the authors mention examples of some of these measures?

Response 9

Thank you very much for your comment. We have included measures to acknowledge and protect healthcare workers. Please see lines 558-559 of the revised manuscript

Reviewer: 3

Overall Comment of Reviewer 3

I congratulate the authors for conducting valuable research work and nicely drafting the manuscript. I thoroughly enjoyed reviewing it. There are a few observation which if authors address would add rigor to the methodology and comprehensiveness of the paper. They are as follows:

Response

Thank you for the comment and we are glad that you enjoyed reading our manuscripts. We have acted upon all the suggestions made.

Comment 1

1) Line 110: The findings of this study will.....The statement cannot be definite. Better to represent a possibility (could/can).

Response 1

Thank you for your comment. The statement has been rephrased to reflect the changes suggested. Please see line 122 of the revised manuscript.

Comment 2

14: Aim-Better would be to also highlight the qualitative nature of work here.

Response 2

Thank you very much for the suggestion. We have added qualitative research to the sentence. Please see line 127 of the revised manuscript.

Comment 3

Data analysis: Authors should provide more information about data interpretation, e.g., triangulation- was it performed or not, Who performed coding and was it rechecked by other investigators or not? Likewise, the categorization of the themes was done by whom? how the consensus was built?

Response 3

Thank you very much for the comment. The data analysis section has been updated to provide specific details on the author's participation in the different stages of the data analysis process. Please see lines 205-214 of the revised manuscript.

Comment 4

Word count should be restricted to 4000, as per the Journals' guidelines. Discussion may be trimmed.

Response 4

Thank you for this observation, we tried to restrict the word count to 4,000 for readability, however, we feel that the extra words are necessary to highlight the interconnection between psychosocial impacts, health system resilience and resilient people and this has been explained in detail in the cover letter to the editor.

VERSION 2 – REVIEW

REVIEWER	Umeokonkwo, Chukwuma Alex Ekwueme Federal University Teaching Hospital Abakaliki, Department of Community Medicine
REVIEW RETURNED	26-Apr-2023

GENERAL COMMENTS	The authors have addressed the concerns raised.
---

REVIEWER	James, Peter University of Sierra Leone College of Medicine and Allied Health Sciences, Pharmaceutical Sciences
REVIEW RETURNED	11-Apr-2023

GENERAL COMMENTS	The authors have attempted to address my comments. However, there are still some comments that they did not or adequately address.
--

	1. . I believe the authors misunderstood my comment regarding paragraph 4 of the introduction based on their response. The authors responded “Thank you for the comment, the fourth paragraph in reference refers to job-related stressors healthcare workers experienced during the Ebola outbreak and not the COVID-19 outbreak. We believe it is important for us to mention such in the manuscript” In my comment, I did not suggest that the references on job-related stressors healthcare workers experienced during the Ebola outbreak were not appropriate, but I believe it equally important to review the available local literature regarding the barriers or challenges healthcare workers experienced that might affect their psychological well-being during the COVID-19 outbreak, given that the fact that the current study is looking at HCW experience during COVID-19 outbreak in Sierra Leone. I suggested the reference because the suggested study reported that HCWs in Sierra Leone considered that their health facilities were ill-prepared to adequately respond to COVID-19 with the majority stating that their facilities lack enough personal protective equipment. In fact, the suggested reference further helps to strengthen the justification for conducting the study (The justification for conducting the study is not clearly stated or discussed) as it only identified the lack enough personal protective equipment as a challenge but failed to establish whether such a challenge directly affected HCWS psychological wellbeing even though it is a potential stressor as it has established during the EBOLA outbreak. I strongly suggest that the authors discuss the local literature regarding HCWs' experience during COVID-19 as potential stressors and use their findings to strengthen their study justification. In addition, to my suggested reference, the authors can consider the following article Parmley LE, Hartsough K, Eleeza O, Bertin A, Sesay B, Njenga A, et al. (2021) COVID-19 preparedness at health facilities and community service points serving people living with HIV in Sierra Leone. PLoS ONE 16(4): e0250236. https://doi.org/10.1371/journal.pone.0250236. 2. The authors were asked to mention the theoretical framework underpinned their study design and if no theoretical framework was used to proffer the reason(s) why. The authors' response was “ An explorative qualitative interview study with data analysed using the framework approach, which supports the generation of themes and provided clear systematic steps while maintaining the transparency of the analytic process²⁵. The study is based on the COnsolidated criteria for REporting Qualitative research (COREQ).” The above response by the authors does not address my comment. I strongly suggest that the authors address my comment. 3. The authors were asked to properly describe the trustworthiness of the study findings by describing how credibility, transferability, confirmability, and dependability were ensured. The authors responded that they have addressed it in the data analysis section and the strength and limitation section. The authors failed to
--	---

	address my comment based on what the authors wrote in these sections. I strongly suggest that the authors create a whole subsection under the methods section and describe how each of the following (credibility, transferability, confirmability, and dependability) was ensured when designing the study and collecting and analysing the data. 4. Currently, it is still unclear how the verification of data sources was done to ensure the accuracy of the translation and completeness of the transcription process. 5, I suggested that the quote come after the summary describing the subtheme. However, in their response, the authors mentioned a quote before the summary sets the scene for the proceedings paragraph, and such a format has been used in published research. Can the authors provide at least five recently published papers in reputable journals that have used such a format in presenting their findings? As a qualitative researcher and a journal editor, most qualitative papers I have read have presented their findings in the form in which the quote comes after the summary describing the subtheme/theme. However, I'm open to learning new presentation formats if it is the new normal and logically justified.
--	---

REVIEWER	Gupta, Snehil AIIMS Bhopal, Department of Psychiatry
REVIEW RETURNED	19-Apr-2023

GENERAL COMMENTS	I congratulate the authors for coming up with the revised manuscript and incorporating suggested changes.
---

VERSION 2 – AUTHOR RESPONSE

Dr. Sia Tengbe (on behalf of all authors)

Reviewer: 2

Dr. Peter James, University of Sierra Leone College of Medicine and Allied Health Sciences,
Southern Cross University

Overall Comment to the Author

The authors have attempted to address my comments. However, there are still some comments that they did not adequately address.

Overall Comment to Reviewer

Dear Dr. Peter James, thank you very much for taking the time to review our manuscript. We have made the necessary revision to the manuscript to address the comments accordingly and detailed these below in italics.

Comment 1

I believe the authors misunderstood my comment regarding paragraph 4 of the introduction based on their response. The authors responded “Thank you for the comment, the fourth paragraph in reference refers to job-related stressors healthcare workers experienced during the Ebola outbreak and not the COVID-19 outbreak. We believe it is important for us to mention such in the manuscript”

In my comment, I did not suggest that the references on job-related stressors

healthcare workers experienced during the Ebola outbreak were not appropriate, but I believe it equally important to review the available local literature regarding the barriers or challenges healthcare workers experienced that might affect their psychological well-being during the COVID-19 outbreak, given that the fact that the current study is looking at HCW experience during COVID-19 outbreak in Sierra Leone. I suggested the reference because the suggested study reported that HCWs in Sierra Leone considered that their health facilities were ill-prepared to adequately respond to COVID-19 with the majority stating that their facilities lack enough personal protective equipment.

In fact, the suggested reference further helps to strengthen the justification for conducting the study (The justification for conducting the study is not clearly stated or discussed) as it only identified the lack enough personal protective equipment as a challenge but failed to establish whether such a challenge directly affected HCWS psychological wellbeing even though it is a potential stressor as it has established during the EBOLA outbreak.

I strongly suggest that the authors discuss the local literature regarding HCWs' experience during COVID-19 as potential stressors and use their findings to strengthen their study justification.

In addition, to my suggested reference, the authors can consider the following article

Parmley LE, Hartsough K, Eleeza O, Bertin A, Sesay B, Njenga A, et al. (2021) COVID-19 preparedness at health facilities and community service points serving people living with HIV in Sierra Leone. PLoS ONE 16(4): e0250236. <https://doi.org/10.1371/journal.pone.0250236>.

Response 1

Thank you very much we have now included this paper and additional other national literature from Sierra Leone (Parmley LE, Hartsough K, Eleeza O, Bertin A, Sesay B, Njenga A, et al. (2021) COVID-19 preparedness at health facilities and community service points serving people living with HIV in Sierra Leone. PLoS ONE 16(4): e0250236. <https://doi.org/10.1371/journal.pone.0250236>) in our introduction to better discuss and situate our findings. We have also strengthened the justification for the paper in the introduction, changing the order to highlight the gaps in understanding and how this paper will feed into policy and practice discussions. Please see lines 119-140.

Comment 2

The authors were asked to mention the theoretical framework underpinned their study design and if no theoretical framework was used to proffer the reason(s) why. The authors' response was

“ An explorative qualitative interview study with data analysed using the framework approach, which supports the generation of themes and provided clear systematic steps while maintaining the transparency of the analytic process. The study is based on the Consolidated criteria for Reporting Qualitative research (COREQ).”

The above response by the authors does not address my comment. I strongly suggest that the authors address my comment.

Response 2

Thank you, in our prior comments we expanded upon the rigour of the explorative qualitative study. We have now expanded this to be clearer that this research is an exploratory qualitative study and also amended the title to show this. Please see line 2.

It was underpinned by the naturalistic paradigm and was primarily inductive (themes emerged from data), although it did draw on some deductive approaches too (from the topic guide and study objectives). Please see lines 146-148; 253-258 which expand on this

We did not use a specific theoretical framework to guide our analysis and interpretation of findings as is common practice within exploratory qualitative research. Rather, we drew on literature pertaining to resilience, trust and social support systems to inform our analytical process and to generate a new theoretical framework related to health worker psychological outcomes in Sierra Leone, as shown within Figure 1. This theoretical framework is further described within our discussion section and we hope is useful to others working in the space for further testing and exploration.

Comment 3

The authors were asked to properly describe the trustworthiness of the study findings by describing how credibility, transferability, confirmability, and dependability were ensured. The authors responded that they have addressed it in the data analysis section and the strength and limitation section. The authors failed to address my comment based on what the authors wrote in these sections. I strongly suggest that the authors create a whole subsection under the methods section and describe how each of the following (credibility, transferability, confirmability, and dependability) was ensured when designing the study and collecting and analyzing the data.

Response 3

We have included this additional sub-section under the methods which includes some of the existing text but with further details to describe how credibility, transferability, confirmability, and dependability was ensured when designing the study and collecting and analysing the data in our study. Thank you very much. Please see lines 262-278.

Comment 4

Currently, it is still unclear how the verification of data sources was done to ensure the accuracy of the translation and completeness of the transcription process.

Response 4

Thank you very much for your comment. We have explained in detail how data source verification was done to ensure the accuracy and completeness of the transcription process in our study. Interviews conducted in krio were listened to and transcribed directly in English by the lead researcher (SMT) who is fluent in both languages. Where there were words or concepts that were hard to understand or directly translate a 3rd opinion was sought and respondent validation undertaken. Please see lines 228-232.

Comment 5

I suggested that the quote come after the summary describing the subtheme. However, in their response, the authors mentioned a quote before the summary sets the scene for the proceedings paragraph, and such a format has been used in published research. Can the authors provide at least five recently published papers in reputable journals that have used such a format in presenting their findings? As a qualitative researcher and a journal editor, most qualitative papers I have read have presented their findings in the form in which the quote comes after the summary describing the subtheme/theme. However, I'm open to learning new presentation formats if it is the new normal and logically justified.

Response 5

Thank you we have reviewed and changed the first quote so that we are consistently first discussing the themes (rather than starting with a quote) and then using illustrative quotes to represent these from different participants. Please see line 302.

VERSION 3 – REVIEW

REVIEWER	James, Peter University of Sierra Leone College of Medicine and Allied Health Sciences, Pharmaceutical Sciences
REVIEW RETURNED	03-Jul-2023

GENERAL COMMENTS	No further comments. The authors have fully addressed my comments.
--

VERSION 3 – AUTHOR RESPONSE